# Protective Role of Selenium-Binding Protein 1 (SELENBP1) in Patients with Ulcerative Colitis

**DOI:** 10.3390/metabo14120662

**Published:** 2024-12-01

**Authors:** Gabriela Fonseca-Camarillo, Janette Furuzawa-Carballeda, Ángel A. Priego-Ranero, Rafael Barreto-Zúñiga, Braulio Martínez-Benítez, Jesús K. Yamamoto-Furusho

**Affiliations:** 1Inflammatory Bowel Disease Clinic, Department of Gastroenterology, Instituto Nacional de Ciencias Médicas y Nutrición, Salvador Zubirán, Mexico City 14080, CP, Mexico; gabrielafaster@gmail.com; 2Department of Immunology, Instituto Nacional de Cardiología, Ignacio Chávez, Mexico City 14080, CP, Mexico; 3Department of Experimental Surgery, Instituto Nacional de Ciencias Médicas y Nutrición Salvador Zubirán, Mexico City 14080, CP, Mexico; jfuruzawa@gmail.com; 4Medicine School, Universidad Panamericana, Mexico City 03920, CP, Mexico; 5Department of Clinical Cardiology, Instituto Nacional de Cardiología, Ignacio Chávez, Mexico City 14080, CP, Mexico; angel.priego@gmail.com; 6Department of Endoscopy, Instituto Nacional de Ciencias Médicas y Nutrición Salvador Zubirán, Mexico City 14080, CP, Mexico; 7Department of Pathology, Instituto Nacional de Ciencias Médicas y Nutrición Salvador Zubirán, Mexico City 14080, CP, Mexico; brauliomb77@yahoo.com.mx

**Keywords:** selenium, selenbp1, ulcerative colitis, remission, inmunoregulation, IBD

## Abstract

Background: The expression of selenium-binding protein 1 (SELENBP1), a molecule responsible for the absorption of selenium in the colon, is crucial for its immunoregulatory effect, but this phenomenon has not been studied in patients with UC. The present study aimed to determine the clinical outcome of SELENBP1 expression in colonic tissue from patients with UC. Methods: The relative mRNA expression of SELENBP1 was analyzed in 34 patients with UC and 20 controls. Statistical analyses were performed with SPSS 19. Results: SELENBP1 gene expression was significantly lower in patients with active UC than those with UC in remission (*p* = 0.003) and within the controls (*p* = 0.04). Overexpression of the SELENBP1 gene was associated with a more benign clinical course characterized by initial activity and more than two years of prolonged remission (OR 23.7, *p* = 0.003) and an intermittent clinical course (OR 47.5, *p* = 0.001), mild histological activity (OR 0.11; 95% CI: 1.00–1.41, *p* = 0.05) and severe histological activity (OR 0.08, 95% CI: 0.008–0.866, *p* = 0.02). SELENBP1-positive cells were found mainly in the submucosa’s inflammatory infiltrate and muscular and adventitia’s internal layers from patients with active UC compared to those in the control group (*p* ≤ 0.001). Conclusions: The upregulation of SELENBP1 was associated with a benign clinical course of UC. This is the first report suggesting the immunoregulatory role of SELENBP1 in patients with UC.

## 1. Introduction

Inflammatory bowel disease (IBD) is a term that encompasses conditions characterized by chronic inflammation of the gastrointestinal tract. It includes ulcerative colitis (UC) and Crohn’s disease (CD). UC is characterized by continuous chronic inflammation of the colonic mucosa and submucosa of the large intestine or colon. At the same time, the CD can occur anywhere between the mouth and the anus but most commonly involves the large and small intestines with healthy parts of the tissue mixed in between inflamed areas, and it has a transmural involvement. Dietary influence has been associated with altered epigenetics, gut microbiota, and immunological factors that trigger UC development [1]. Many studies have examined the connections between diet, etiology, signs, symptoms, and IBD [2,3,4].

IBD patients experience malnutrition, undernutrition, or even overnutrition, mainly caused by suboptimal nutritional intake, alterations in nutrient requirements and metabolism, malabsorption, and excessive gastrointestinal losses. A suboptimal nutritional status, anemia, and low micronutrient (iron, vitamins A, B1, B2, and D, zinc, selenium, etc.) serum levels impair both the induction and maintenance of remission and the quality of life of IBD patients [5].

The micronutrient selenium is mainly obtained from bread and cereals, meat, fish, eggs, and milk/dairy products. In northern Mexico, a previous study reported selenium content in fish and seafood. In the daily Mexican diet, beans (16 μg/100 g), corn (14 μg/100 g), wheat (18 μg/100 g), corn tortilla (16 μg/100 g), flour tortilla (25 μg/100 g), Brazil nut (approx. 2000 μg/100 g), avocado (11.73 μg/100 g), cauliflower (6.57 μg/100 g), lettuce (6.62 μg/100 g), chili (2.76 μg/100 g) and watermelon (9.05 μg/100 g) are the most significant selenium contributors [6,7]. The selenium consumed in foods and supplements exists in several organic and inorganic forms, including selenomethionine (plant and animal sources and supplements), selenocysteine (animal-based foods, e.g., pork, beef, mutton, chicken, fish, milk, and egg are humans’ best sources), selenate, and selenite (mainly supplements). The main route for selenium intake is via the diet, whereas the contribution from water and air is negligible. The average recommended daily intake of selenium for adults is 53 µg per day for women and 60 µg per day for men [8].

Selenium is relevant in regulating the immune response, inflammatory processes, and oxidative stress. Considering this, their low serum concentration may exacerbate inflammation through epithelial barrier dysfunction, altered mucosal immunity, and increased production of proinflammatory cytokines [9].

In this vein, the low selenium serum concentration is thought to be a risk factor for several chronic diseases associated with oxidative stress and inflammation, including CD [10,11,12].

Evidence on selenium supplementation in IBD patients under treatment with Infliximab demonstrated that adding the micronutrient to Infliximab can reduce adverse drug reactions and improve clinical symptoms in mild to moderate UC patients [13].

Selenium’s biological actions occur through low-molecular-weight metabolites and selenoproteins. In humans, 25 selenoproteins have been identified, including selenium-binding protein 1 (SELENBP1), a highly conserved protein [14].

SELENBP1 mRNA is ubiquitously expressed in humans, with the highest expression occurring in the adult kidney, duodenum, liver, spleen, lung, brain, and blood. The SELENBP1 gene encodes a binding carrier protein for selenocysteine (an active form of selenium) that catalyzes the oxidation of methanethiol, an organosulfur compound produced by gut microbiota. Two of the reaction products of methanethiol oxidation, hydrogen peroxide, and hydrogen sulfide, serve as signaling molecules for colonocyte differentiation. SELENBP1 may play essential roles in several fundamental physiological functions, including protein degradation, intra-Golgi transport, cell differentiation, cellular motility, redox modulation, and the metabolism of sulfur-containing molecules [15].

A previous study aimed to identify biomarkers of intestinal repair by large-scale screening of multiple transcriptomic and scRNA-seq datasets of patients with inflammatory bowel disease (IBD) and identified 10 marker genes that potentially contribute to intestinal barrier repairing and SELENBP1. This evidence suggests the protective role of SELENBP1 in IBD; expression of this healing marker was specific to absorptive cell types in the intestinal epithelium. Migration of cells along the colonic crypt-luminal axis was also associated with SELENBP1 induction [16].

The SELENBP1 protein responsible for the absorption/transport of selenium has not been studied in patients with UC.

This study aimed to determine the gene expression and localization of SELENBP1 in immune cells in patients with UC and its association with clinical outcomes.

## 2. Materials and Methods

This study included 34 patients with UC (17 active and 17 in remission) and 20 controls without histological evidence of colitis or neoplasia. Clinical activity was analyzed by the novel integral index for UC or Yamamoto-Furusho index [17], which includes variables in the disease index were the number of bowel movements per day; biochemical values (calprotectin, albumin, hemoglobin, high-sensitivity C-reactive protein, leukocytes, and platelets); and endoscopic and histologic findings (measured through the subscales of the Mayo and Riley scores, respectively). All subjects were recruited at the Inflammatory Bowel Disease Clinic at Instituto Nacional de Ciencias Médicas y Nutrición Salvador Zubirán between May 2015 and May 2019.

The control group consisted of rectal mucosal samples of individuals without colonic inflammation who underwent colonoscopy for anemia. The absence of intestinal inflammation was verified histologically. The sample size was not estimated because this is an exploratory study, and we included the highest number of patients compared to previous studies regarding gene expression in UC patients.

### 2.1. Sample Processing and Gene Expression Analysis

Rectal mucosal biopsies were taken via colonoscopy, placed in 1000 μL of RNAlater^®^ (R0901, Sigma–Aldrich, St Louis, MO, USA), and stored at −80 °C until processing. The methodology was performed similarly to previous studies [18,19,20,21]. The colonic tissue was disrupted, homogenized in lysis/binding buffer, and purified. The purified RNA was then diluted in 100 μL of elution buffer. Two hundred nanograms of total RNA was reverse transcribed into cDNA with a random hexamer primer (Roche Diagnostics, Mannheim, Germany) [18,19,20,21]. For relative mRNA expression analysis, we performed the previous standardized methodology used by Fonseca Camarillo G. et al. [20,21]. PCR amplification of the SELENBP1 RNA transcripts was performed with 20 ng of cDNA and a Probe Master Mix Kit in a volume of 10 µL. PCRs were run in a Light Cycler 480; Roche Diagnostics, Mannheim, Germany for 45 cycles.

The housekeeping gene glycerin aldehyde 3-phosphate dehydrogenase (GAPDH) was employed for relative quantification. The following primers were used for quantification: SELENBP1 (Gene Bank: NM_003944.2); LEFT: attgttaagggaggccctgt; RIGHT: gctctgggactttagtt; UPL67; GADPH (Gene Bank: NM_002046.3): forward: gcccaatacgaccaaatcc; and reverse: agccacatcgctcagaca UPL: 60. The relative quantification of the target gene mRNA was conducted using LightCycler software 4.1 according to the 2-delta-delta Ct method. For the q-PCR assay, the linearity and reproducibility were evaluated (V.C. < 10%).

### 2.2. Co-Localization of SELENBP1 in Patients with UC and Controls

To detect the SELENBP1 protein, we employed tissue samples from 15 patients’ refractory to conventional treatment with severe disease and 10 control patients without intestinal inflammation.

To determine the localization of SELENBP1 with CD16+ cells (CD16 is a marker of neutrophils, monocytes, and macrophages), a double-staining procedure was carried out with a Multiview (mouse-HRP/rabbit-AP) immunohistochemistry kit (Enzo Life Sciences, Inc., Farmingdale, NY, USA).

For this method, 50 μL of the mouse monoclonal IgG2b anti-SELENBP1 antibody (cat sc-373726; SANTACRUZ) and C-terminal polyclonal rabbit IgG1 anti-CD16+ (Santa Cruz Biotechnology, Santa Cruz, CA, USA) were used.

We followed the methodology of the double-staining procedure devised by Furuzawa-Carballeda et al. [22,23]. Sections were deparaffinized and rehydrated through xylene and graded alcohol solutions. Enzyme antigen retrieval was carried out for 2 min (Enzo Life Sciences, Inc., Farmingdale, NY, USA), and endogenous peroxidase activity was blocked in the tissue with 3% H_2_O_2_ in methanol. Nonspecific background staining was avoided with an immunohistochemistry serum-free background-blocking solution (Enzo Life Sciences). A mixture of mouse monoclonal and rabbit polyclonal antibodies at 10 μg/mL was incubated for 40 min at room temperature in a wet chamber. The slides were washed with wash buffer and then incubated with PolyView IHCh reagent (anti-mouse-HRP, Enzo Life Sciences, Inc.) and PolyView IHCh reagent (anti-rabbit-AP, Enzo Life Sciences, Inc.) for 20 min. The SELENBP1 antigen was visualized using horseradish peroxidase (HRP)/3,3′-diaminobenzidine (DAB), and the CD16+ was visualized with alkaline phosphatase (AP)/Permanent Red. Intestinal tissues were counterstained with Mayer’s hematoxylin and mounted in an aqueous mounting medium. Negative control staining was performed with the universal negative control reagent designed to work with rabbit, mouse, and goat antibodies (IHCh universal negative control reagent; Enzo Life Sciences, Inc.) and with normal human serum diluted 1:100 instead of primary antibodies. The controls were subjected to nonspecific staining and endogenous enzymatic activities.

Blinded research performed a morphometric evaluation of the immunostained sections in a light microscope. In brief, SELENBP1-expressing CD16 cells were counted in at least three optical fields from each slide at high-power magnification (320×). The average values per slide were used for statistical analysis. The results are expressed as the mean ± standard error of the cell mean (SEM) and were quantified by the program Image-Pro Plus version 5.1.1.

### 2.3. Statistical Analysis

Statistical analysis was performed using GraphPad Prism version 6 and SPSS version 19. Descriptive statistics were performed, and categorical variables were compared using Fisher’s exact test. A one-way analysis of rank variance Kruskal–Wallis was performed, if the Kruskal–Wallis test was significant, a post hoc analysis (Dunn’s test) was performed for pairwise comparisons and comparisons versus a control group. Kruskal–Wallis and Spearman correlation tests were used. A strong correlation was defined as a Spearman coefficient between ±0.50 and ±1, a medium correlation between ±0.30 and ±0.49, and a weak correlation below 0.29. Fisher’s exact test and odds ratio (OR) were used to assess the significance of associations between SELENBP1 gene expression and clinical features. For SELENBP1/CD16-immunoreactive cells, the Tukey test was used for all pairwise comparisons of the mean ranks of the groups. Data are expressed as the median, range, and mean ± standard deviation (SD)/standard error of the mean (SEM). *p* values less than or equal to 0.05 were considered significant.

### 2.4. Ethical Declarations

The present study was conducted following the ethical standard of the Helsinki Declaration, the Good Clinical Practice guidelines, and local regulatory requirements. The institutional review board authorized the study, each patient provided written informed consent, and all patient information, including illustrations, was anonymized.

## 3. Results

### 3.1. Patient Demographic, Clinical, and Biochemical Characteristics

In this study, 54 patients were included: 34 with UC (17 with active UC and 17 in remission) and 20 without intestinal inflammation. The patient’s demographic, clinical, and biochemical characteristics are shown in Table 1 and Table 2.

Regarding systemic inflammation in patients with active UC, platelets and leukocytes were found at significantly higher levels than controls (Figure 1B,C).

Fetal calprotectin, a very sensitive marker for inflammation in the gastrointestinal tract and a biomarker for diagnosis, monitoring disease activity, treatment guidance, and prediction of disease relapse, was increased at statistically significant levels in patients with active disease compared to those with inactive disease (Figure 1F). Hemoglobin and hs-CRP did not show differences between the study groups (Figure 1A,D).

### 3.2. SELENBP1 Gene Expression Is Decreased in Active UC Compared to Remission Disease

SELENBP1 gene expression was significantly lower in patients with active UC than in patients with UC in remission (*p* = 0.003) and within the control group (*p* = 0.04; Figure 2). Similar SELENBP1 gene expression in UC patients with non-inflamed tissue was associated with a more benign clinical course characterized by initial activity and more than two years of prolonged remission (OR 23.7, *p* = 0.003) and an intermittent clinical course (OR 47.5, *p* = 0.001). Additionally, low mRNA levels of SELENBP1 correlated significantly with mild histological activity (OR 0.11; 95% CI: 1.00–1.41, *p* = 0.05) and severe histological activity (OR 0.08, 95% CI: 0.008–0.866, *p* = 0.02). Histopathological characteristics were defined by Riley score [24] Mildly active is defined by neutrophil infiltration of <50% of sampled crypts or cross-sections, no ulcers or erosions and severely active is defined by erosion or ulceration, distortion of the crypts and pseudovillus appearance of the colon surface.

### 3.3. Intestinal Production of SELENBP1 Is Increased in All Layers in the Intestine of Severe UC Patients Compared to Controls

SELENBP1 is a cytoplasmic and nuclear (nucleoli) protein expressed high in the intestine distal enterocytes. Nonetheless, it has been characterized in the plasma membrane of monocytes/macrophages. SELENBP1/CD16 double-positive cells represent a population of immune cells, such as NK cells or M1 macrophages, in the inflammatory infiltrate of the submucosa and internal layers muscular and adventitia from patients with active UC compared to those in the control group (*p* ≤ 0.001; Figure 3).

## 4. Discussion

Selenium in the human body mainly protects against oxidative damage, regulating immune function and inhibiting inflammatory response through selenoproteins. Selenocysteine is incorporated into the amino-acid sequence of selenoproteins during translation, coded for by a UGA codon in the messenger ribonucleic acid (mRNA) coding region. All selenoproteins contain selenocysteine, the 21st amino acid, within their active sites. When the selenium (Se) status is affected by inflammation and hypoxia, SELENBP1 (SBP1, hSP56) contributes to disease-specific micronutrient metabolism [8].

SELENBP1 is encoded by a gene located at 1q21.3 near the epidermal differentiation complex, which is closely related to the terminal differentiation of the human epidermis and contains genes that encode the S100A family members (associated with tumor progression and metastasis). Unlike the proteins containing selenocysteine or selenomethionine, SELENBP1 directly binds selenium through cysteine 57, converting it to glycine and reducing the half-life of the protein, inducing mitochondrial damage, and attenuating the degree of phosphorylation of signaling protein such as p53 and GSK3β compared to the native protein expressed at similar levels [25]. SELENBP1 is a phylogenetically conserved protein across different species and is not induced by dietary selenium exposure, as shown in rodents and the nematode C. elegans. It is ubiquitously expressed and subcellularly localized in the nucleus and the cytoplasm. Intracellularly, SELENBP1 affects redox homeostasis by reciprocal interference with the antioxidative activity of the selenoenzyme glutathione peroxidase 1 (GPx-1) [26]. Downregulation of SELENBP1 expression has been associated with carcinogenesis and the poor prognosis of various human malignancies [27,28].

Moreover, a study by Huang et al. demonstrated that the decreased expression of SELENBP1 could lead to a higher GPX1 activity and reduced HIF-1α expression in hepatocellular carcinoma. Typically, the high level of oxidative stress in cancer cells (usually caused by tumor microenvironment or drug-induced ROS) would lead to cellular apoptosis rather than survival or transformation due to the inhibition of GPx-1 activity, but not its expression, by the upregulated SELENBP1. The translocation of GPX1 to the nucleus in cancer cells under oxidative stress may facilitate the antioxidant functions of GPx-1. In contrast, the formation and combination of GPx-1 and SBP1 nuclear bodies might inhibit this process. However, as the expression of SELENBP1 in HCC and many other cancers is reduced, it promotes tumor malignant transformation and even metastasis (increasing cell motility, promoting cell proliferation, and inhibiting apoptosis only under oxidative stress), shorter overall survival periods, and higher rates of disease recurrence [27].

SELENBP1 activates transcriptional induction of p21 through a p53-independent mechanism. The p21 protein, also known as p21^WAF1/CIP1^, is encoded by the *CDKN1A* gene mapped to chromosome 6p21.2 and belongs to the Cip/Kip family of CDK inhibitors. Many studies have demonstrated p21 as a multifunctional, broad-acting protein that plays key roles in cell cycle regulation, migration, and apoptosis. Thus, p21 is intricately regulated via p53-dependent and -independent pathways. SELENBP1 inhibits the phosphorylation of c-Jun and STAT1, which belong to the AP-1 and STAT families. The combined suppression of c-Jun and STAT1 activities may be essential for SELENBP1-mediated transcriptional induction of the p21 protein. C-Jun has been previously shown to interact with STAT3 and co-operatively regulate the transcription of their target genes in bladder cancer [28].

On the other hand, a gene set enrichment analysis showed that the SELENBP1 gene was significantly enriched in several pathways, such as programmed death 1 (PD-1) signaling, signaling by interleukins, TCR signaling, collagen degradation, MHC class II antigen presentation, co-stimulation by the CD28 family, and antigen processing cross-presentation. Moreover, SELENBP1 positively correlated with eosinophils, B cells, and Th17 cells, and negatively correlated with macrophages, Th1 cells, neutrophils, Th2 cells, Tgd, NK cells, T helper cells, Tem, cytotoxic cells, Tcm, CD8 T cells, aDC and DC in patients with colorectal cancer [29].

This is the first study to show the differential gene expression and co-localization of selenium-binding protein 1 (SELENBP) in intestinal tissue from patients with UC under different clinical conditions (active and remission).

The upregulation of the SELENBP1 gene was found in patients with remission UC. This gene was associated with a more benign clinical course characterized by initial activity and more than two years of prolonged remission. Conversely, decreased SELENBP1 gene expression was associated with mild histological activity and a severe and intermittent clinical course.

These findings suggest the immunoregulatory or protective role of the SELENBP1 gene in patients with UC, where high expression of SELENBP-1 can induce or maintain disease remission probably by negatively regulating the cascade of inflammatory mediators. At the same time, a decrease in its level is associated with frequent and severe relapses.

Additionally, we detected double-positive SELENBP1+/CD16+ cells among inflammatory infiltrates of the mucosa, submucosa, muscular space, and adventitia from patients with active UC.

This is the first demonstration of co-localization of SELENBP1+/CD16+ in intestinal tissue. Our findings are interesting because the primary source of these immune cells was in the submucosal layers of patients with active UC. In the mucosal layer, we detected a lower percentage of positive cells, which may suggest that the immune cells responsible for selenium absorption were not found in the mucosa, which is the main site of involvement and effect in patients with UC.

We measured markedly lower SELENBP1 mRNA levels in active UC patients than in non-inflamed UC patients and controls. Our data showing the induction of SELENBP1 expression during remission suggest that selenium’s immunoregulatory role and biological actions occur through selenoproteins.

SELENBP1 is a methanethiol oxidase (MTO) related to the MTO of methylotrophic bacteria that converts methanethiol to H_2_O_2_, formaldehyde, and H_2_S, an activity unknown to humans [10].

However, the physiological role of SELENBP1 has long remained elusive. Evidence shows its involvement in intracellular protein degradation and transport [30,31]. SELENBP1 is a marker of terminally differentiated epithelial cells in the colon and may act as a tumor suppressor [31,32,33,34].

SELENBP1 has been reported to be ubiquitously expressed, with the highest levels occurring in the liver, kidneys, and intestine [34].

Epidemiological studies have suggested an inverse association between selenium levels and inflammatory bowel disease (IBD), which includes Crohn’s disease and ulcerative colitis and can potentially progress to colon cancer.

However, the underlying mechanisms are not well understood.

Kudva Ak et al. (2015) suggested that changes in the cellular oxidative state coupled with altered expression of selenoproteins in macrophages drive the switch from a proinflammatory to an anti-inflammatory phenotype to efficiently resolve inflammation in the gut and restore epithelial barrier integrity [35]. The specific implications of SELENBP1 in the gastrointestinal tract have not been explored.

This study revealed high protein expression in all intestinal layers of UC patients, which appears to be related to disease activity.

This transversal study is limited to the Mexican population. Since diet is related to the function and expression of selenoproteins, further studies in other populations with more individuals and a pertinent characterization of dietary patterns are needed.

Moreover, translational studies in the UC are needed since there is some evidence that SELENBP1 could play a role in the pathogenesis of intestinal inflammation in murine models and human IBD.

## 5. Conclusions

The upregulation of the SELENBP1 gene was associated with a more benign clinical course characterized by initial activity and more than two years of prolonged remission. Downregulation of SELENBP1 was associated with an intermittent clinical course of mild and severe histological activity. This is the first report on the intestinal production of SELENBP1 by immune cells in patients with UC suggesting the immunoregulatory role of selenoproteins in UC.

## Figures and Tables

**Figure 1 metabolites-14-00662-f001:**
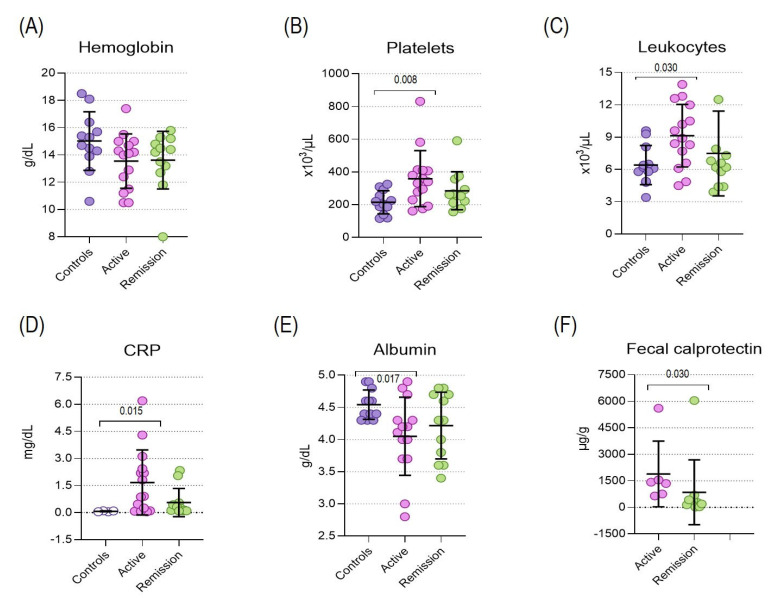
Systemic inflammation in UC patients and controls. (**A**) Hemoglobin, (**B**) Platelets, (**C**) Leukocytes, (**D**) hs-CRP, (**E**) Albumin, and (**F**) Fecal calprotectin. Bars represent mean ± standard deviation.

**Figure 2 metabolites-14-00662-f002:**
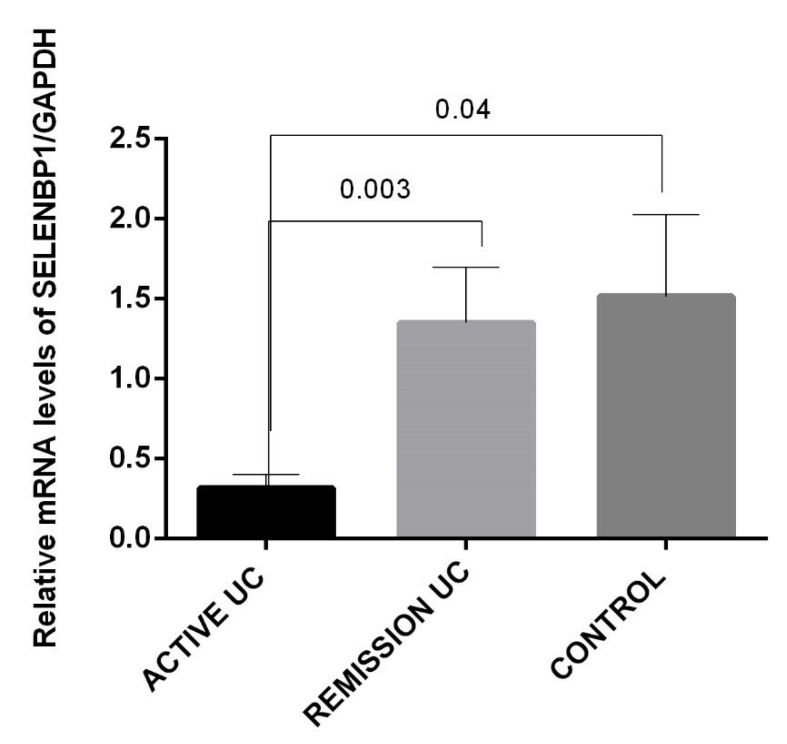
Relative mRNA levels of SELENBP1 in patients with active UC and controls. The gene expression of SELENBP1 was significantly lower in patients with active UC than in patients in the remission group (*p* = 0.003) and controls (*p* = 0.04). The bars show the means ± standard errors of the means of the SELENBP1 transcript levels. The Kruskal–Wallis test was used to assess differences among groups.

**Figure 3 metabolites-14-00662-f003:**
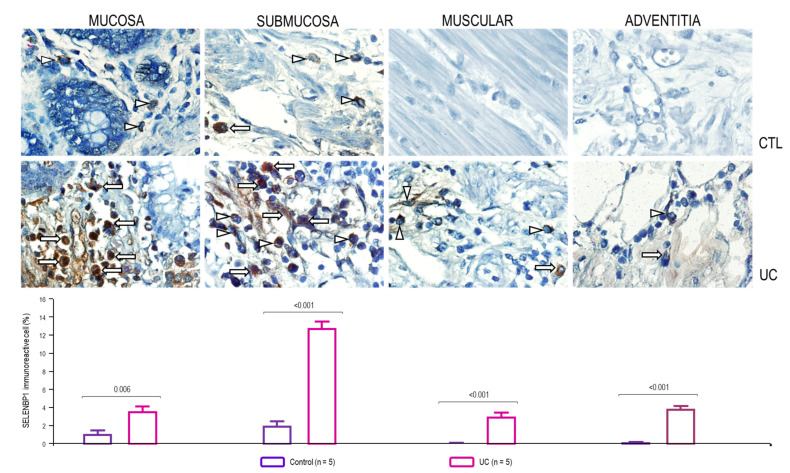
Representative immunoperoxidase analysis of SELENBP1 in patients with active UC and controls. CTL: control group, no inflamed colonic tissue (n = 10; **upper panel**) and UC: active ulcerative colitis tissue (n = 10, **lower panel**). The arrows indicate CD16+ (red cells)/SELENBP1+ (brown cells) double-positive immunoreactive cells (burgundy) in the mucosa, submucosa, muscular space, and adventitia. Arrowheads depict single positive cells. The original magnification was 600×. The bars indicate the results of immunohistochemistry analysis of CD16/SELENBP1-expressing cells in non-inflamed colonic tissues (control, n = 5) and tissues from active UC patients.

**Table 1 metabolites-14-00662-t001:** Demographic characteristics of active and remission UC patients and controls.

Clinical Characteristics	Active UC (n = 17)	Remission UC (n = 17)	Controls (n = 20)
Gender			
Male, n (%)	8 (47)	8 (47)	8 (40)
Female, n (%)	9 (53)	9 (53)	12 (60)
Age			
Median (range)	35.5 (24–58)	40 (22–80)	47.4 (20–67)
Disease evolution			
Median (range)	7.8 (1–24)	8 (1–27)	not applicable
Extent of Disease			
E1- proctitis, n (%)	4 (24)	3 (18)	not applicable
E2- Left-sided, n (%)	3 (18)	1 (6)	
E3: Pancolitis, n (%)	10 (58)	13 (18)	
Extraintestinal manifestations			
Present, n (%)	6 (35)	6 (35)	not applicable
Absent, n (%)	11 (65)	11 (65)	
Clinical Course of Disease			
Initially, n (%)	5 (29)	11 (65)	not applicable
Intermittent, n (%)	12 (71)	6 (35)	
Continuous, n (%)	0 (0)	0 (0)	
Medical Treatment			
5-Aminosalicylates, n (%)	15 (88)	14 (82)	not applicable
Steroids, n (%)	6 (35)	3 (18)	
Thiopurines, n (%)	6 (35)	3 (18)	
Anti-TNFα, n (%)	2 (12)	0 (0)	

**Table 2 metabolites-14-00662-t002:** Laboratory variables of patients with UC and controls.

Laboratory Values	Active UC (n = 17)	Remission UC (n = 17)	Controls (n = 20)
Hemoglobin (g/dL)			
Mean ± SD	13.63 ± 1.87	13.78 ± 1.77	15.01 ± 2.14
Median	14.2	14.2	15
Range	10.50–17.4	8–15.8	10.6–18.50
Leukocytes (×10^3^/µL)			
Mean ± SD	9.04 ± 3.26	7.07 ± 3.3	6.40 ± 1.81
Median	8.4	6.10	6.15
Range	4.5–15.10	3.0–17.8	3.4–9.6
Platelets units (×10^3^/µL)			
Mean ± SD	349 ± 162	283.05 ± 96.09	214.66 ± 70.81
Median	331	229	211
Range	161–831	68–436	116–325
Erythrocyte Sedimentation Rate (mm/h)			
Mean ± SD	24.8 ± 21.9	13.1 ± 12.0	3.5 ± 1.9
Median	24.7	11.0	3.0
Range	2.0–66.0	2.0–46.0	2.0–6.0
High-sensitivity C-reactive protein (mg/dL)			
Mean ± SD	1.67 ± 1.80	0.54 ± 0.78	0.21 ± 0.23
Median	0.90	0.19	0.11
Range	0.05–6.19	0.02–2.33	0.05–0.56
Albumin (blood serum; g/dL)			
Mean ± SD	4.17 ± 0.70	4.53 ± 0.66	4.54 ± 0.22
Median	4.2	4.7	4.5
Range	2.80–5.30	3.40–5.30	4.3–4.9
Fecal calprotectin (μg/g)			
Mean ± SD	1621.1 ± 1839	888 ± 1744.83	Not determined
Median	1351	282	
Range	0–5605	20–6044	

## Data Availability

Research data are available upon reasonable request.

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
