# Peer review of "Protective Role of Selenium-Binding Protein 1 (SELENBP1) in Patients with Ulcerative Colitis"

_metabolites, 2024, doi:10.3390/metabo14120662_

Round 1

Reviewer 1 Report

Comments and Suggestions for Authors
  1. This study evaluated the association of SELENBP1 with a benign clinical course of UC. Although the conclusions is novelty, but is not consistent with the evidence and arguments presented and the authors did not address the main question posed. The references and Table/Figures are not appropriate.

  2. the results were not well presented, such as P=0.05 OR=0.11, IC=1.00-1.41 in Introduction
  3. the sample size was too small
  4. the systemic inflammation in UC patients and controls was not appropriate, no difference was found in CRP?
  5. anthough SELENBP1 was detected in CPR and  immunoperoxidase, the conclusion was not enough to support.

Author Response

Reviewer 1

Dear Reviewer

We appreciate your comments and review of our manuscript

Comments and Suggestions for Authors

  1. This study evaluated the association of SELENBP1 with a benign clinical course of UC. Although the conclusions is novelty, but is not consistent with the evidence and arguments presented and the authors did not address the main question posed. The references and Table/Figures are not appropriate.

R: We consider that the findings suggest the immunoregulatory or protective role of the SELENBP1 in patients with UC, where its high expression can induce or maintain disease remission probably by negatively regulating the cascade of inflammatory mediators. At the same time, a decrease in its level is associated with frequent and severe relapses.

We are aware that this can only be confirmed with functional studies being carried out and that, for the moment, we will not integrate into this manuscript but into a subsequent one.

At the reviewer's suggestion, tables and figures were adapted to the journal format.

  1. The results were not well presented, such as P=0.05 OR=0.11, IC=1.00-1.41 in Introduction.

R: Thanks to the reviewer, we have amended the errors.

  1. The sample size was too small

R: We agree that the sample size is small. However, we detected significant differences in SELENBP1 expression between the groups.

  1. The systemic inflammation in UC patients and controls was not appropriate, no difference was found in CRP?

R: Thank you very much for the observation. When reviewing the control group, we had an out layer of 4.6 mg/dl. We eliminated the subject to avoid losing statistical significance between the control group and the group of patients with active UC.

  1. Although SELENBP1 was detected in CPR and immunoperoxidase, the conclusion was not enough to support.

R: We recognize the fact that functional studies are required to be able to categorically conclude the participation of SELENBP1 in the regulation of inflammation. However, these are currently being carried out and will not be included in this manuscript for the moment but in a subsequent one.

Reviewer 2 Report

Comments and Suggestions for Authors

1) Methods: The relative mRNA expression of SELENBP1 was analyzed in 34 patients with UC and 20 24 controls. Statistical analyses were performed with SPSS 19. Expand the methodology a bit more. 

2) Thus, due to the bowel damage and chronic inflammatory status of the IBD course. Do not start sentence with .... due to.

3) SELENBP1 mRNA is ubiquitously expressed in humans, with the highest expression 82 occurring in the adult kidney, duodenum, liver, spleen, lung, brain, blood, etc. Do not put etc. 

4) The relative SELENBP1 gene expression, such as the molecule responsible for the 98 absorption/transport of selenium, has not been studied in patients with UC. Clarify the sentence. 

5) Correct the typo error. visualized with alkaline phosphatase (A.P.)/

6) Rectal mucosal biopsies were taken via colonoscopy, placed in 1 ml of RNA (R0901, 119 Sigma‒Aldrich), and stored at -80°C until processing. Correct the unit. 

7) Ethical declarations can be more clearly explained. 

8) Intestinal production of SELENBP1 is increased in all layers in the intestine of se- 250 vere UC patients compared to controls. Heading should not be like descriptive sentence. 

9) SELENBP1 activates transcriptional induction of p21 through a p53-independent 296 mechanism. Cite a reference here.

10) These findings suggest the immunoregulatory or protective role of the SELENBP1 323 gene in patients with UC. Integrate the line with other statements of the paragraphs. 

Author Response

Title: " Protective role of selenium-binding protein 1 (SELENBP1) in patients with Ulcerative Colitis ".

Dear Editors and Referees,

We appreciate your comments concerning our manuscript. These observations are all valuable and helpful for revising and improving the paper and have an important guiding significance to research. We have reviewed the comments carefully and made the appropriate corrections.

Reviewer 2

Comments and Suggestions for Authors

  • Methods: The relative mRNA expression of SELENBP1 was analyzed in 34 patients with UC and 20 controls. Statistical analyses were performed with SPSS 19. Expand the methodology a bit more. 

As suggested by the reviewer, the methods section was modified and improved.

"Statistical analysis was performed using GraphPad Prism version 6 and SPSS version 19. Descriptive statistics were performed, and categorical variables were compared using Fisher's exact test. A one-way analysis of rank variance Kruskal‒Wallis, if Kruskal‒Wallis test was significant, a post hoc analysis (Dunn's test)  was performed for pairwise comparisons and comparisons versus a control group. Kruskal‒Wallis and Spearman correlation tests were used. A strong correlation was defined as a Spearman coefficient between ±0.50 and ±1, a medium correlation between ±0.30 and ±0.49, and a weak correlation below 0.29. Fisher's exact test and odds ratio (OR) were used to assess the significance of associations between SELENBP1 gene expression and clinical features. For SELENBP1/CD16-immunoreactive cells, the Tukey test was used for all pairwise comparisons of the mean ranks of the groups. Data are expressed as the median, range, and mean ± standard deviation (SD)/standard error of the mean (SEM). P values less than or equal to 0.05 were considered significant."

  • Thus, due to the bowel damage and chronic inflammatory status of the IBD course. Do not start sentence with .... due to.

As suggested by the reviewer, the sentence was depleted.

  • SELENBP1 mRNA is ubiquitously expressed in humans, with the highest expression occurring in the adult kidney, duodenum, liver, spleen, lung, brain, blood, etc. Do not put etc. 

Thanks to the reviewer, we have amended the word. Line 87

  • The relative SELENBP1 gene expression, such as the molecule responsible for the absorption/transport of selenium, has not been studied in patients with UC. Clarify the sentence. 

Thanks to the reviewer, the statement was modified. Line 102

  • Correct the typo error. visualized with alkaline phosphatase (A.P.)/

Thanks to the reviewer, we have amended the typo error. Line 163

  • Rectal mucosal biopsies were taken via colonoscopy, placed in 1 ml of RNA (R0901, 119 Sigma‒Aldrich), and stored at -80°C until processing. Correct the unit.

We have amended the units to 1000 µl. Line 123

  • Ethical declarations can be more clearly explained. 

As suggested by the reviewer, the ethical declarations paragraph was more explained. "The present study was conducted following the ethical standard of the Helsinki Declaration, the Good Clinical Practice guidelines, and local regulatory requirements. The institutional review board authorized the study, each patient provided written informed consent, and all patient information, including illustrations, was anonymized."

  • Intestinal production of SELENBP1 is increased in all layers in the intestine of severe UC patients compared to controls. Heading should not be like descriptive sentence. 

The heading was modified. Line 254

  • SELENBP1 activates transcriptional induction of p21 through a p53-independent Cite a reference here.

Reference 28 was incorporated.

  • These findings suggest the immunoregulatory or protective role of the SELENBP1 gene in patients with UC. Integrate the line with other statements of the paragraphs. 

As suggested by the reviewer, this paragraph was added:

These findings suggest the immunoregulatory or protective role of the SELENBP1 gene in patients with UC, where high expression of SELENBP-1 can induce or maintain disease remission probably by negatively regulating the cascade of inflammatory mediators. At the same time, a decrease in its level is associated with frequent and severe relapses.

Reviewer 3 Report

Comments and Suggestions for Authors

Dear Authors.

The research described in the article is extremely interesting and comprehensive. The purpose of the research is to identify the links between selenium entering the body and the suppression of one or another type of cancer. It was found that the up regulation of the SELENBP1 gene was associated with a more benign clinical course characterized by initial activity and more than two years of prolonged remission. Downregulation of SELENBP1 was associated with an intermittent clinical course of mild and severe histological activity. This is the first report on the intestinal production of SELENBP1 by immune cells in patients with ulcerative colitis, suggesting the immunoregulatory role of selenoproteins in ulcerative colitis.

The research and description are done at a very high level and are undoubtedly worthy of publication in such a respected Journal.

However, I would like to note a number of comments.

1. Authors write: The decreased expression of SELENBP1 could lead to a higher GPX1 activity and reduced HIF-1alpha expression in hepatocellular carcinoma, indicating that SELENBP1 might exert its tumor suppressive function as a regulator of the tumor redox microenvironment.

I would like to see a more detailed explanation of this idea in the article and what the authors of the article think about this matter.

2. Authors write: Our  data showing the induction of SELENBP1 expression during remission suggest that selenium's immunoregulatory role and biological actions occur through selenoproteins. 

A more detailed explanation of this scientific conclusion is required.

3. Authors write: Since diet is related to the function and expression of selenoproteins, further studies in other populations with more individuals and pertinent characterization of dietary patterns are needed. 

It is necessary to show in the introduction the sources of replenishment of selenium in food in the Mexican population. It is necessary to cite a number of plants with a high content of selenium, used in Mexican national dishes. In contrast, cite two or three articles - studies on selenium in plants growing on the Eurasian continent and accordingly used in cooking.

Author Response

Title: " Protective role of selenium-binding protein 1 (SELENBP1) in patients with Ulcerative Colitis ".

Dear Editors and Referees,

We appreciate your comments concerning our manuscript. These observations are all valuable and helpful for revising and improving the paper and have an important guiding significance to research. We have reviewed the comments carefully and made the appropriate corrections.

Reviewer 3

Comments and Suggestions for Authors

Dear Authors.

The research described in the article is extremely interesting and comprehensive. The purpose of the research is to identify the links between selenium entering the body and the suppression of one or another type of cancer. It was found that the up regulation of the SELENBP1 gene was associated with a more benign clinical course characterized by initial activity and more than two years of prolonged remission. Downregulation of SELENBP1 was associated with an intermittent clinical course of mild and severe histological activity. This is the first report on the intestinal production of SELENBP1 by immune cells in patients with ulcerative colitis, suggesting the immunoregulatory role of selenoproteins in ulcerative colitis.

The research and description are done at a very high level and are undoubtedly worthy of publication in such a respected Journal.

However, I would like to note a number of comments.

  1. Authors write: The decreased expression of SELENBP1 could lead to a higher GPX1 activity and reduced HIF-1alpha expression in hepatocellular carcinoma, indicating that SELENBP1 might exert its tumor suppressive function as a regulator of the tumor redox microenvironment.

I would like to see a more detailed explanation of this idea in the article and what the authors of the article think about this matter.

Typically, the high level of oxidative stress in cancer cells (usually caused by tumor microenvironment or drug-induced ROS) would lead to cellular apoptosis rather than survival or transformation due to the inhibition of GPX1 activity, but not its expression, by the upregulated SBP1. The translocation of GPX1 to the nucleus in cancer cells under oxidative stress may facilitate the antioxidant functions of GPX1. In contrast, the formation and combination of GPX1 and SBP1 nuclear bodies might inhibit this process. However, as the expression of SBP1 in HCC and many other cancers is reduced, it promotes tumor malignant transformation and even metastasis (increasing cell motility, promoting cell proliferation, and inhibiting apoptosis only under oxidative stress), shorter overall survival periods, and higher rates of disease recurrence.

  1. Authors write: Our data showing the induction of SELENBP1 expression during remission suggest that selenium's immunoregulatory role and biological actions occur through selenoproteins. 

A more detailed explanation of this scientific conclusion is required.

As suggested by the reviewer, this paragraph was added:

These findings suggest the immunoregulatory or protective role of the SELENBP1 gene in patients with UC, where high expression of SELENBP-1 can induce or maintain disease remission probably by negatively regulating the cascade of inflammatory mediators. At the same time, a decrease in its level is associated with frequent and severe relapses.

  1. Authors write: Since diet is related to the function and expression of selenoproteins, further studies in other populations with more individuals and pertinent characterization of dietary patterns are needed. 

It is necessary to show in the introduction the sources of replenishment of selenium in food in the Mexican population. It is necessary to cite a number of plants with a high content of selenium, used in Mexican national dishes. In contrast, cite two or three articles - studies on selenium in plants growing on the Eurasian continent and accordingly used in cooking.

According to a study, the total selenium intake of the Mexican population appears to be adequate. The study also found that the plasma values of selenium in children compare favorably with other reported values, indicating an adequate selenium status in the population. 

Some foods that are important sources of selenium in the Mexican diet include: Beans, Corn tortillas, and Flour tortillas.

A previous study reported selenium content in selected food products in northern Mexico, including fish and seafood. In the daily Mexican diet, beans (16 mg/100g), corn (14 mg/100g), wheat (18 mg/100g), corn tortilla (16 mg/100g), fluor tortilla (25 mg/100g), brazil nut (approx 2000 mg/100g), avocado (11.73 mg/100g), cauliflower (6.57 mg/100g), lettuce (6.62 mg/100g), chilli (2.76 mg/100g) and watermelon (9.05 mg/100g) are the most significant Se contributors.

  1. Wyatt, C. J., Meléndez, J. M., Acuña, N., & Rascon, A. (1996). Selenium (Se) in foods in northern Mexico, their contribution to the daily Se intake and the relationship of Se plasma levels and glutathione peroxidase activity. Nutrition Research, 16(6), 949–960. doi:10.1016/0271-5317(96)00094-2 
  2. Antioxidants and Natural Compounds in Mexican Foods

Written By José Luis Silencio Barrita, Sara Montaño Benavides and Santiago Sánchez Submitted: 09 February 2015 Reviewed: 24 September 2015 Published: 11 November 2015 DOI: 10.5772/61626

Round 2

Reviewer 1 Report

Comments and Suggestions for Authors

no additional comment